# Integrating nurses' experiences with supporting behaviour change for cardiovascular prevention into a self-management internet platform in Finland and the Netherlands: a qualitative study

Cathrien RL Beishuizen,[1] Ulrika Akenine,[2] Mariagnese Barbera,[3] Anna Rosenberg,[3] Mandana Fallah Pour,[2,4] Edo Richard,[5,6] Hilkka Soininen,[3] Francesca Mangialasche,[7] Miia Kivipelto,[3,7] A Jeannette Pols,[8] Eric Moll van Charante[1]

For numbered affiliations see end of article.

**Correspondence to**
Cathrien RL Beishuizen;
c.r.beishuizen@amc.uva.nl

## ABSTRACT

**Objectives** Global ageing is linked to an increased burden of cardiovascular disease and dementia, which calls for better prevention strategies. Self-management and eHealth applications are regarded as promising strategies to support prevention. The aim of this study was to explore nurses' best practices concerning behaviour change guidance for cardiovascular (CV) prevention in order to learn how to optimally integrate them into a coach-supported internet platform for CV self-management.

**Design** Qualitative focus group study in Finland and the Netherlands. Discussions were audiotaped and transcribed. Data were thematically analysed following principles of grounded theory.

**Setting** Dutch and Finnish primary care settings.

**Participants** Six Finnish and seven Dutch primary care nurses with experience in CV prevention.

**Results** Similar best practices were found in both countries and comprised of (1) establishing a relationship of trust, (2) managing awareness and expectations and (3) appropriate timing and monitoring of the process of behaviour change. However, the Finnish and Dutch nurses used different approaches for accomplishment of these practices, which was reflected in their recommendations for online support. Both groups emphasised that online support should be combined with human support and integrated into regular care. Finnish nurses had more confidence in patient self-management and remote communication than Dutch nurses, who emphasised the importance of face-to-face contact and preferred to keep control of medical aspects of prevention.

**Conclusions** Differences in Dutch and Finnish's nurses' practices for supporting CV prevention appear to reflect their local healthcare practices, which should be taken into account when designing internet platforms for health self-management. Including cognitive health as a goal of CV prevention might stimulate motivation for health behaviour change.

**Trial registration number** ISRCTN48151589; Pre-results.

### Strengths and limitations of this study

► This international focus group study directly compares best practices of Finnish and Dutch primary care nurses in cardiovascular (CV) prevention.

► Language barriers were overcome by closely aligning our research methodologies, multiple iterations in the analysis and extensive meetings between the research teams.

► Our original approach, following grounded theory, enabled us to learn from the best practices of nurse experts in 'traditional' face-to-face CV preventive care and integrate these into recommendations for optimal support for health behaviour change through novel eHealth applications.

► Since our samples and number of focus groups were limited, we are aware that our findings are not exhaustive, especially since we found that local healthcare practices substantially impacted the study results.

► Due to the exploratory nature of our study, our findings are preliminary and might be influenced by differences in nurses' clinical experience and in the ages of their patient populations.

## INTRODUCTION

Global ageing places an increasing demand on healthcare systems, partially due to the absolute rise in cardiovascular disease (CVD) and dementia cases.[1,2] As these disorders share a number of risk factors, effective cardiovascular (CV) prevention could also lead to the prevention of dementia.[3–6] CV prevention requires health behaviour change, the process of 'initiating and maintaining behaviours that reduce health risks and control existent chronic

disease'.[7] In CV prevention, core behaviours consist of a healthy lifestyle (healthy diet, sufficient physical activity and non-smoking) and adherence to medication. Although the processes supporting health behaviour change have been theorised extensively,[8–12] putting them into practice remains a challenge[13 14] and novel, more effective, approaches are needed.[15] Two strategies of current interest are self-management and eHealth. In self-management, the individual, instead of the healthcare professional, takes the lead in the management of his/her risk factors, and therefore in behaviour change.[16 17] eHealth applications are attractive because of their wide reach and their potential to support self-management because of their suitability for health education, interactivity and monitoring.[18 19] Although researchers and policymakers have high expectations of eHealth and self-management, more intensive face-to-face interventions still achieve better results than eHealth applications.[20] To learn how self-management and behaviour change could be best stimulated and maintained online, we consulted 'experts in the field', that is, nurses experienced in health behaviour change in the context of CV prevention.

This project is part of the Healthy Ageing Through Internet Counselling in the Elderly (HATICE) study, which includes a European randomised controlled trial (ISRCTN48151589) testing a coach-supported internet platform for self-management of CV risk factors in older people to prevent CVD and cognitive decline.[21] In an international focus group study, we aimed to (1) explore nurses' best practices concerning behaviour change guidance for CV prevention, including the potential for dementia prevention, and (2) learn how to integrate their practices into a coach-supported internet platform (the online-support setting). This study took place in Finland and the Netherlands, two of the three countries that participated in the HATICE study. Since the HATICE project aims to develop an internet platform, that is, implementable across all European healthcare systems, we also explored the influence of local healthcare practices.

## METHODS
### Design
We performed an international qualitative focus group study following principles of grounded theory.[22 23] The Consolidated Criteria for Reporting Qualitative Research (COREQ)-checklist is included for full methodological information (online supplementary appendix 1).[24]

### Participants and setting
For sampling, we followed the grounded theory methodology of studying a healthcare practice by consulting field experts.[25] In this perspective, Finnish and Dutch primary care nurses experienced in CV preventive care were considered most eligible for this study and selective purposive samples were obtained. In Finland, occupational healthcare nurses were recruited because of their important role in preventive CV care (please see

table 1 for a description of Finnish and Dutch primary healthcare systems and the position of Finnish occupational health). Fourteen nurses working in a semi-private healthcare centre in Kuopio (Eastern Finland) were invited by email and telephone and six female nurses (43%) consented to participate. Being occupational health nurses, they cared mostly for patients of working age. Duration of clinical experience with CV prevention ranged from 2 to 35 years. In the Netherlands, we recruited primary care nurses experienced in CV risk management. A group of 32 nurses experienced in CV preventive care working in general practices in two urban areas in the centre of the Netherlands was invited by email and telephone. Seven female nurses (22%) consented to participate. The unanimous reason for non-participation by Finnish and Dutch nurses was lack of time. The Dutch participating nurses cared for patients of all ages. Duration of clinical experience with CV prevention ranged from 3 to 11 years. Table 2 contains further characteristics of the nurses.

The study was presented to the medical ethics committee of the Academic Medical Centre in the Netherlands and a waiver was provided. In Finland, neither application for ethical approval nor a waiver was required. All participants provided written informed consent.

### Data collection
We conducted one focus group in the Netherlands (autumn 2013) and one in Finland (December 2015). We regarded focus groups as the most appropriate method to answer our research aims, because they enabled us to explore the experiences and attitudes of the nurses as thoroughly as possible, as the nurses could directly respond to each other's opinions and develop their ideas through the discussion. In each country, an experienced focus group moderator chaired the sessions, while an assistant-moderator noted non-verbal communication and summarised the discussions. The discussion was conducted using a topic list as a flexible guide (box 1 and online supplementary appendix 2). After the Dutch session, the topic list was refined for the Finnish focus group, to further explore the following topics: the nurse–patient relationship, attitudes towards eHealth applications, shaping optimal online support, the role of the nurse in this versus the role of the patient and dementia prevention. The moderators asked open-ended questions, following principles of grounded theory, to inductively approach the data. Both moderators first asked the nurses about their activities in CV prevention and how they supported their patients in the process of behaviour change (*Part 1* of the focus group). The Finnish moderator also asked the nurses about their experiences with prevention of dementia. In *Part 2* of the focus group, the HATICE internet platform was presented (box 2, a full description of the platform is reported elsewhere[26]) and the nurses were asked how they would optimally support their patients in an online setting. Both sessions lasted ~2 hours. The discussions were audiorecorded and transcribed.

**Table 1** The Finnish and Dutch primary care systems

| | Finland | The Netherlands |
|---|---|---|
| Organisation | Primary care is delivered through public healthcare centres and through occupational health facilities.[37–39] Companies offer occupational health facilities to their employees, including both preventive and curative health services, delivered through semiprivate healthcare centres that work with nurses in a similar fashion to the public centres. | General practices or small healthcare centres. In most general practices, continuity of care is ensured by allocating the patient to one (GP). |
| Main focus | Primary care has a strong position and important gatekeeper function. Health promotion and disease prevention have been the main focus of healthcare policy for decades. | Access to care for everyone and solidarity through medical insurance.[40 41] GPs are gatekeepers of the healthcare system and provide acute, chronic and preventive care. |
| Accessibility | Often, healthcare centres cover large geographical areas that are sparsely populated and often have staff shortages, contributing to long waiting lists and lack of personal continuity of care. Due to these waiting lists, many employees go to their occupational health service instead. | Since the Netherlands is densely populated, people often live within a short distance of their general practice. Access to GPs is efficient; there are no waiting lists.[42] |
| Role of primary care nurses | Important position: nurses work in close collaboration with GPs and have their own consulting hours to assess patients. Regarding cardiovascular prevention, they monitor patients with diabetes, hypertension and dyslipidaemia, as described in national guidelines.[43–45] | Important position: for several decades, GPs have delegated tasks to practice nurses in chronic disease management. Currently, nurses provide a substantial part of cardiovascular risk management care, including diabetes care, following regional and national guidelines and work descriptions.[9 46–48] |
| Patient autonomy and eHealth culture to date | The first European country to introduce a law (in 1993) defining the patient's right to access to all medical information and the right to autonomy (patient's informed consent for any medical treatment). All healthcare centres use electronic medical records. A national patient data repository is under development to provide patients complete access to their own electronic medical record.[49] | Informed consent is ensured by law, but in daily practice, consent is often assumed and only explicitly discussed when treatment options can have far-reaching consequences.[41] Almost all GPs use electronic medical records. Patients have the right to inspect their medical records, but do not have complete access to them. |

GP, general practitioner.

## Coding and analysis

In each country, two researchers coded and thematically analysed the transcripts following principles of grounded theory.[22 23] Themes were derived inductively from the data and were not hypothesised beforehand. Open coding and identification of initial themes was first performed by the two researchers independently. Thereafter, codes and themes were compared. Dissimilarities were discussed until consensus was reached. Initial theme structure was then discussed with the senior researchers involved. In Finland, since the researchers were not Finnish native speakers, the transcript was translated into English and

**Table 2** Characteristics of the participating Finnish and Dutch nurses

| No | Country | Age | Training | Type of CVD prevention | Internet use at work |
|---|---|---|---|---|---|
| 1 | FI | 55 | Occupational health nurse | Prim/sec prev | Email, guideline use, referral, patient contact |
| 2 | FI | 42 | Occupational health nurse | Prim/sec prev | Email, guideline use, referral |
| 3 | FI | 25 | Occupational health nurse | Prim/sec prev | Email, guideline use, referral, patient contact |
| 4 | FI | 45 | Occupational health nurse | Prim/sec prev | Email, guideline use, referral, patient contact |
| 5 | FI | 49 | Occupational health nurse | Prim/sec prev | Email, guideline use, referral, patient contact |
| 6 | FI | 60 | Occupational health nurse | Prim/sec prev | Guideline use, patient contact |
| 1 | NL | 43 | General nurse, practice nurse* | Prim/sec prev | Email, guideline use, referral, patient contact |
| 2 | NL | 49 | Practice nurse | Prim/sec prev | Email, guideline use, referral, patient contact |
| 3 | NL | 51 | Practice nurse | Prim prev | Email, guideline use, referral, patient contact |
| 4 | NL | 53 | General nurse, practice nurse | Prim/sec prev | Email, guideline use, referral |
| 5 | NL | 42 | Practice nurse | Sec prev | Email, guideline use, referral, patient contact |
| 6 | NL | 45 | General nurse, practice nurse† | Prim/sec prev | Email, guideline use, referral, patient contact |
| 7 | NL | 65 | General nurse, practice nurse | Prim/sec prev | Email, guideline use, referral, patient contact |

*Practice nurse: received specific nursing training to work in general practice.
†General nurse: received general nursing training to work as a hospital-based general nurse.
CVD, cardiovascular disease; FI, Finland; NL, the Netherlands; prev, prevention; prim, primary; sec, secondary.

**Box 1    Main topics discussed**

**Part 1**
► Prevention of cardiovascular disease and dementia: attitude and experiences.
► Good guidance of behaviour change.
► Relationship with the patient.

**Part 2**
► Attitude towards the internet platform and online support.
► Role and responsibilities of the internet coach.
► Interaction with the patients online.

cross-checked by the Finnish focus group moderator, who was a Finnish native and fluent English speaker. In this way, the complete analysis of the Finnish data could be performed in English. After the initial analysis performed locally, themes and corresponding quotations of the Dutch sessions were also translated into English. The two research teams then had two meetings to discuss the structure of main themes and categories. The analysis phase[22 23] was an iterative process, during which the researchers from both teams repeatedly returned to their data files to add, merge and refine themes, until a definite theme structure was agreed on by all authors. During the iterative analysis phase, the researchers discussed the themes and alternatives, and it was proposed that the local healthcare context could influence the differences found between caring styles of the two groups of nurses. Therefore, the research teams introduced their local healthcare systems (box 2) to each other and these insights were used in further interpretation of the findings. A summary of the final conclusions was returned to the participants for feedback.

### Patient and public involvement
Patients were not involved in the design of this substudy of HATICE. However, patients were involved in the development of the HATICE eHealth application by means of focus groups with the projected target population of the HATICE eHealth application and by means of consulting patient organisations (Dutch Heart Foundation and Dutch and Finnish Alzheimer Association).[26] Results of this substudy were disseminated to the participants by means of a written summary.

**Box 2    Key features of the HATICE internet platform with coaching**

► Patient-centred: the patient can login on to a personal portal to review and manage his/her personal cardiovascular (CV) risk profile.
► Improving health knowledge: the patient can access educational modules about CV risk factors and lifestyle.
► Goal setting and self-monitoring: the patient can set his/her own goals for behaviour change and monitor how he/she is doing by entering self-measurements or keeping a diary.
► Coaching: the coach monitors the patient's self-management and they can communicate online through messages.

## RESULTS
We analysed the data from *Part 1* (the nurses' experiences and practices with supporting the process of behaviour change for CV prevention, including the potential for dementia prevention) (Part 1) and *Part 2* (the nurses' suggestions on how to integrate their experiences in an online-support setting, stimulated by a demonstration of the HATICE platform[26]) together, jointly informing the identification of three main themes. The themes can be understood as the nurses' preconditions for effective behaviour change guidance in their patients: establishing a relationship of trust; awareness and expectation management; and appropriate timing and monitoring. These were regarded as being equally important in 'off-line' and 'on-line' healthcare. Below, they are reported separately in relation to *Parts 1 and 2*, to distinguish the nurses' clinical experiences and practices in current healthcare settings from their recommendations for optimal online support.

### Part 1: nurses' experiences and practices with supporting the process of behaviour change for CV prevention
#### Preconditions for effective behaviour change guidance
*Establishing a relationship of trust*
According to both the Finnish and Dutch nurses, the basis of behaviour change support lay in establishing a relationship of trust with the patient, that is, developing a good nurse–patient relationship over time, in which the individual felt at ease and respected and comfortable enough to open up about lifestyle and behaviour issues:

> For lifestyle change, for prevention, a relationship based on mutual trust is pivotal. It is good to have long-standing contact with people. Then you know what is going on in someone's life and in that, some kind of trust will grow, so that people really start believing what you are saying to them. And then, over time, people will start practising healthy behaviours that maybe they had no intention to follow, in the beginning. (Dutch nurse 1)

The nurses reported personalising and tailoring their support to each patient as skills used in order to stimulate trust. For example:

> And you need to get a good picture of the situation, so that you don't give the same information to everyone. That's of no use. You need to think what the central issues are for this patient. What are the things he or she seems to have resources for? What are the goals that the client sets? What is the client able to do, and with what kind of intensity? What will the time span be like? And I also ask my client directly what kind of support he or she would like. I try to offer what the client thinks he or she needs. (Finnish nurse 1)

Interestingly, when further exploring these skills, the nurses expressed different preferences regarding the ideal mode of communication. The Dutch nurses emphasised the importance of repeated face-to-face contact and

inperson continuity to establish a good relationship. For Finnish nurses, an initial face-to-face contact only seemed sufficient to establish a working relationship:

> And here [refers to the HATICE platform] the initial contact and information session at the beginning is very important because I guess a sort of a relationship needs to be established here as well. In the same way. There are still people behind this platform. (Finnish nurse 1)

Thereafter, they were comfortable with further phone or email contact and did not regard this as less personal than face-to-face contact. Email contact was also considered to have advantages:

> But sometimes this kind of online communication could be less complicated…than face to face.(Finnish nurse 5)

> I have noticed in my work that some people prefer contacting me by e-mail and not by phone. [Others agree] On the phone they might think that they are disturbing me or that it's bad timing, but one can write an e-mail or something anytime. (Finnish nurse 3)

### Awareness and expectation management

A second precondition was awareness and expectation management: checking the patients' level of knowledge and expectations regarding prevention and personal CV risk. Nurses thought that most patients had considerable knowledge of CVD prevention, especially in Finland, due to a long-standing tradition of community-based CV prevention (the North Karelia project[27]). Nonetheless, both groups of nurses had experienced that people were not especially aware of their personal CV risk status:

> That's it, isn't it. For many people, their health is not a concern yet. You can list them the facts, and they hear and read it everywhere, that it is unhealthy to be overweight and that they need to exercise more, but right now, they are not yet bothered by it.(Dutch nurse 6)

Because of this lack of a sense of urgency, the nurses regarded the ability to educate their patients about the consequences of health behaviours as an essential skill of their profession. Once awareness and motivation had grown, people often had unrealistic expectations and the nurses needed to act as *'myth busters'* (Finnish nurse 4):

> And when we are, however, not able to offer the magic pills or wonder tricks, the clients may sometimes be disappointed when all I can suggest is these boring methods: diet and physical activity. And we cannot offer them a magic solution. (Finnish nurse 4)

Often, once people were motivated to change their health behaviours, they also tended to set unrealistic goals, which the nurses then needed to bring back to realistic proportions:

> Start small. Do not make it too big. If you are obese, many people do not like it to go to the gym, they think the gym is only for lovely slim figures. You cannot convince them that that's not true. Therefore, it is important: try things first yourself. What can you do with small steps at home by yourself, before going outside? You have to start liking exercise.(Dutch nurse 3)

Lastly, the nurses actively prepared their patients for failures during the process of behaviour change, as these were seen as inevitable:

> I usually tell the patients that they're allowed to fail; but even so, they are invited to, and they should come to the appointments. So then we can check the situation again, and set a new goal if needed.(Finnish nurse 1)

With the Finnish nurses, coaching on CV risk was also related to the potential for dementia prevention. They suggested that many patients feared dementia and lacked knowledge about the disease and treatment and prevention options, creating a stigma towards this condition. The nurses were aware of the link between CVD and dementia, but felt they lacked sufficient knowledge and training to provide proper support:

> Well, we have not had knowledge of the reasons for dementia for that long. And these connections haven't been…the research is recent: well, at least more recent than the research about heart diseases. (Finnish nurse 5)

They found that educating patients on the link between CVD and dementia would be a good starting point to raise awareness. Potentially, this could enhance motivation for CV prevention:

> What is good for the heart—and we know what's good for the heart—is also good for the brain, but not everyone knows this. I think this link would be good to be aware of: you protect your heart but also the most important part of your body which is the brain. (Finnish nurse 4)

### Appropriate timing and monitoring

The third precondition mentioned by the nurses was appropriate timing and monitoring: providing professional support at appropriate times and monitoring the progress of the patient towards behaviour change. Regular follow-up appointments stimulated adherence and motivation:

> After three months, your plan fades away, your goal, your motivation. (Dutch nurse 3).

> …that there is a possibility for follow-up. Usually it motivates people when someone looks after you: how are you progressing, no matter if the target is, for example, smoking cessation or increasing physical activity. (Finnish nurse 5)

Monitoring ensured that the nurses could support their patients when they experienced obstacles or failures, although this could be difficult:

Disappointments also play a role. For example: a guy with diabetes, he quit smoking but then his sugar levels went up and he needed to start with insulin. How do you explain that [to him]? Well, I challenge you to keep his attitude up and to maintain his motivation. (Dutch nurse 5)

When discussing monitoring lifestyle behaviours, both nurse groups attributed themselves a supportive role, putting the patient in charge, because lifestyle was seen as the patient's personal domain. However, regarding the medical components of preventive care (control of hypertension, diabetes and hypercholesterolemia), the Dutch nurses attributed a more directive role to themselves and the medical practice to avoid mistakes and complications. In contrast, the Finnish nurses regarded their patients as capable of staying in charge and described themselves as mentors:

It is also one of the nurse's responsibilities to be a contact person, support and a sort of mentor, and also to refer the patient to a doctor if the nurse notices that something is going wrong. (Finnish nurse 4)

### Part 2: integrating the nurses' strategies into an online-support setting
#### Establishing a relationship of trust
All of the nurses regarded the presence of a coach as being essential for guaranteeing personal support. The Finnish nurses felt that online coaching could successfully establish a relationship of trust, provided that the coach was a real person:

Because of this social interaction on this website [the HATICE platform], the participant has a familiar and friendly person [as a coach] and not just some distant virtual coach who is a stranger. […] it's good that this combines the real-life person with the online contact, maybe it feels more comfortable and familiar [for the participant]. (Finnish nurse 4)

An initial face-to-face consultation with the patient could strengthen the establishment of a good relationship. Overall, for the Finnish nurses, online support was an obvious step forward in innovating healthcare:

Well at least I think that this is absolutely the trend [others nod and agree], that all the services will be at least partly available online for the patients. Partly like this [via internet] and partly with human contact. I think that it's an inevitable part of the future. (Finnish nurse 1)

In contrast, the Dutch nurses could not imagine the platform and coach fully substituting their personal guidance:

The strength of our guidance is the personal contact we have with the patients. […] that enables us to give them some subtle support and give them a small push in the right direction. To delegate all of that to an online coach just like that, that seems difficult to me. Then all personal contact will disappear. (Dutch nurse 7)

#### Awareness and expectation management
All nurses regarded the internet platform a suitable means to raise awareness and increase health literacy. Managing expectations related to online support was considered very important, because misunderstandings could arise more easily through this method. Therefore, the coach should explain what could be expected from the platform and their support:

Communication is very important in the beginning: what it is we do, and what do they expect from the goals. (Dutch nurse 1)

#### Appropriate timing and monitoring
The nurses envisioned that online, the patient would be in charge of timing of support and monitoring of progress. The coach would have a reactive role, providing support in response to the patient's demand. However, the nurses felt the coach also needed to be proactive, in case people showed signs of losing motivation. This would require insight into people's activities on the platform:

[…] the nurse can also see it [the diary] and check. If the participant fails to achieve the goals, the nurse can go back and check what might have been the problem. (Finnish nurse 5)

Both groups thought the platform should be aligned to regular healthcare. The Finnish nurses envisioned that the online coach could work in the same fashion as the nurses currently did, targeting both lifestyle and medical components of their patient's health. The Dutch nurses stressed that not everybody would be able to self-manage, and therefore would be in need of different intensities of coaching:

I think 2 or 3 types of platform users will arise: people who really get the concept of self-management (and start coaching themselves), people who need the coach (and give the coach access to their complete profile) and a group in-between, alerting the coach if a goal has not been met. (Dutch nurse 2)

While discussing this topic, they expressed ambivalence as to whether it was 'safe' to entrust their patients with self-management when it related to medical issues:

I tend to think: if it is self-management, you shouldn't want to get yourself involved in that [medication use], you should leave that to the GP. On the other hand, if someone's blood pressure is constantly rising, then you do want to know which medication someone is

taking, to get the complete picture. Because then you check whether there might be a problem in medication-use. (Dutch nurse 3)

At the end of this discussion, for safety reasons, they concluded that they preferred a platform focusing on lifestyle only, leaving medical issues within the control of the GP practice.

## DISCUSSION
### Principal findings and interpretation
In this international focus group study, we identified three main themes that both the Finnish and Dutch nurses emphasised as the most important preconditions for effective behaviour change support in CV prevention, and potentially, prevention of cognitive decline: (1) establishing a relationship of trust, (2) managing awareness and expectations and (3) appropriate timing and monitoring of the process of behaviour change. These preconditions were also regarded as important for providing optimal online support. The nurses stressed that a coach providing human support, and integration with regular care, were essential elements to achieve this. They expressed, however, different ideas on their implementation (figure 1).

As mentioned in the Introduction section, making and maintaining health behaviour change is notoriously complex. This was confirmed by the nurses we interviewed, but their clinical experience provided us with clear preconditions for optimal behaviour change

support. The nurses used slightly different approaches to achieve these preconditions, both in their current practice and in their ideas regarding online support. To establish a relationship of trust, the Dutch nurses relied more on face-to-face contact than the Finnish nurses, which appeared to make them more sceptical about the effectiveness of online coaching. The Finnish nurses took a mainly supportive role in monitoring, whereas Dutch nurses emphasised a more directive role for themselves and the general practice, with regard to medical aspects of preventive guidance. This different attitude towards patient autonomy is of interest and may be influenced by different factors, including healthcare culture, geographical factors, nurse factors and patient factors. Regarding healthcare culture, as described in box 2, table 1 although aims of preventive care are very similar between Finland and the Netherlands, patient empowerment and patient autonomy have received more emphasis in Finland than in the Netherlands. The nurses' ideas about their own responsibilities and patient autonomy may be aligned with the way patient autonomy is being shaped in the two healthcare systems as well as with the description of nurses' responsibilities in local CV risk management guidelines. The different attitudes about face-to-face contact can be further understood from a geographical perspective. Finland is a large but very sparsely populated country and the Netherlands is a very small but densely populated country. This has influenced current organisation and accessibility of care, and probably also attitudes

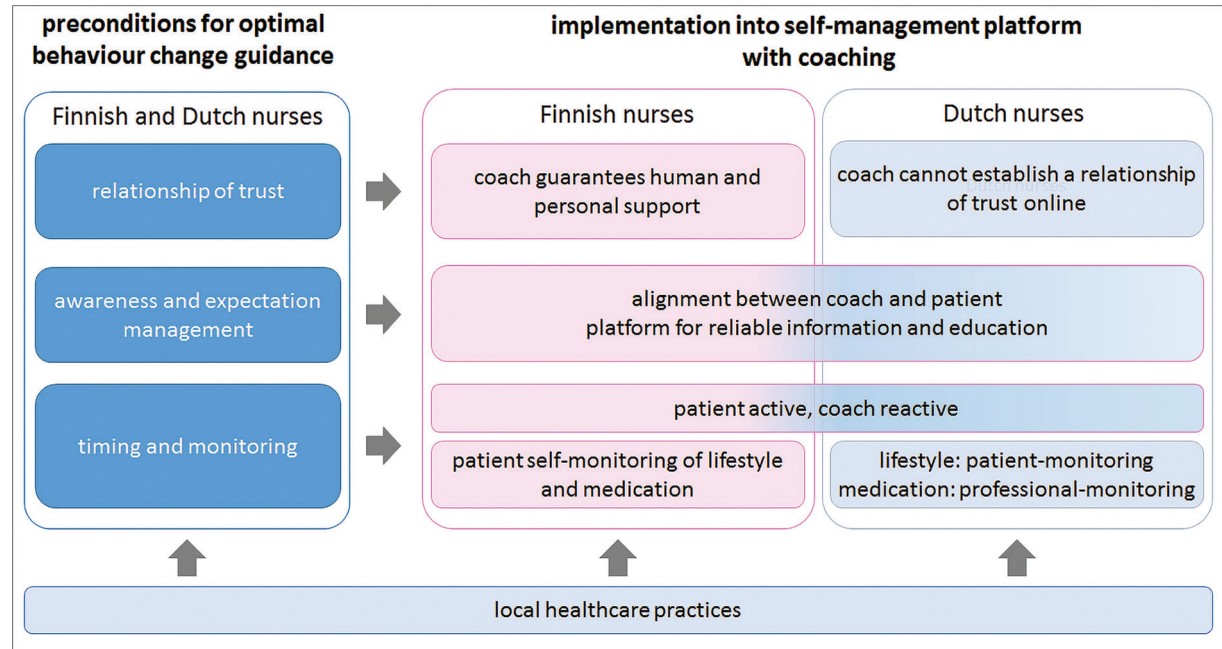

**Figure 1** Schematic visualisation of the main themes and their connections. Left, the three main preconditions for good behaviour change guidance in cardiovascular preventive care that both Finnish and Dutch nurses identified, are depicted. Right of this, it is shown how the Finnish and Dutch nurses suggest to realise these preconditions in the online setting. Since there were differences between the nurses this is depicted separately for the Finnish and Dutch nurses. Below it is shown that local healthcare practices influenced both the preconditions and their operationalisation (not shown in figure but explained in results section) and the integration into online support.

towards care (see box 2) table 1. In this perspective, the step towards online care is likely to be smaller for the Finnish nurses. Last, the differences in the ages of the patient populations may also have influenced our findings, since the nurses might see more potential for eHealth applications in younger patients, who they might regard as being more autonomous in their health behaviours.

Our results concerning dementia prevention are very preliminary, but of special interest. The Finnish nurses liked the idea of including cognitive health as a goal for CV preventive care, as dementia was regarded as a growing public health problem, and a combined approach could increase people's motivation to engage in behaviour change. However, the nurses felt they could not provide proper support, given their limited knowledge and training on one hand, and the limited extent of conclusive scientific evidence on the other.

### Strengths and limitations

The HATICE project is novel in its aim to develop a generic innovative CV prevention strategy for older people that can be used across European healthcare systems, especially since it involves eHealth. Our qualitative research design enabled us to use the best practices of nurse experts in 'traditional' face-to-face CV preventive care to make recommendations for optimal health behaviour change support through novel internet platforms. In qualitative research, international joint analyses are not common because of language barriers. To overcome these, we put much effort into the alignment of our research methodology. The frequent interactions and extensive meetings of the research teams enabled us to explore our findings in the context of the local healthcare systems. Following grounded theory methodology,[25] we deliberately selected nurses who we regarded experts in preventive CV care. Our research also has some limitations that may have influenced our findings. Information on non-participation was limited. The Finnish nurses had, on average, more years of clinical experience with CV prevention than the Dutch nurses. The patient populations of the Finnish and Dutch nurses were not identical with respect to age. Both of these factors may have influenced our findings. However, since the clinical experience of both groups was very similar, and both countries have similar aims for CV prevention, we deem the selection of these nurses appropriate for our research purpose. Since we only performed two focus groups, we cannot exclude that a wider range of views could have been collected. For example, one might expect that themes related to training and education requirements would have emerged more prominently from the discussions, but this issue only was mentioned with regard to cognitive health. A further limitation is that cognitive health was only discussed with the Finnish nurses. This issue should be elaborated further in future studies. The striking similarities in the principal themes found in both countries and the consistency of our findings with

previous literature mitigates fears that our samples were too limited. Last, when reviewing a summary of our findings, the nurses confirmed that their experiences and views were reflected and did not add new ones, emphasising that the most relevant themes were captured.

### Comparison with existing literature

The importance of a relationship of trust, clarifying patients expectations and providing personally tailored support where also main themes in other European qualitative studies on CV preventive care with nurses or patients.[28–32] The positive attitude of the Finnish nurses on self-management of medical issues was consistent with another Finnish study about nurses' and physicians' perceptions of patients' responsibilities in self-care.[33] The reserved attitude of the Dutch nurses was also reflected in a survey among Dutch healthcare professionals, where 50% feared that patients' direct access to their medical record would cause misunderstandings and unnecessary anxiety.[34] A recent qualitative systematic review on nurses' experiences of facilitators and barriers of using telehealth also reported both positive and negative attitudes of nurses towards telehealth. With regard to the nurse–patient relationship, nurses mentioned, on the positive side, that telehealth could improve trusting relationships and lower access to care. On the negative side, nurses reported that telehealth could lead to a loss of human contact. Differences in attitudes were not linked to local healthcare cultures.[35] Finally, the conviction of all nurses in our study that a coach was essential to complement the internet platform, is supported by a meta-analysis we performed showing that internet interventions combined with human support were more effective than 'stand-alone' interventions.[36]

### Implications for practice

Finnish and Dutch nurses have similar experiences with and views on supporting behaviour change for CV prevention, but use different practical approaches towards their patients. Including the maintenance of cognitive health as a goal of CV prevention can provide novel opportunities to frame health behaviour change for both prevention of dementia and CVD, and might augment people's motivation for prevention, but this suggestion should be studied further. The nurses' experiences provide valuable directions for shaping online support in internet platforms for CV self-management. This study also indicates that, when introducing new forms of preventive healthcare that involve patient self-management, like internet platforms, local healthcare practices are to be taken into account to achieve optimal engagement.

### Author affiliations

[1]Department of General Practice, Amsterdam UMC/University of Amsterdam, Amsterdam, The Netherlands
[2]Department of Neurobiology, Care Sciences and Society, Division of Clinical Geriatrics, Karolinska Institutet and Karolinska University Hospital, Stockholm, Sweden

[3]Department of Clinical Medicine/Neurology, University of Eastern Finland, Helsinki, Finland

[4]Department of Neurobiology, Care Sciences and Society, Division of Occupational Therapy, Karolinska Institutet, Huddinge, Sweden

[5]Department of Neurology, Amsterdam UMC/University of Amsterdam, Amsterdam, The Netherlands

[6]Department of Neurology, Donders Institute for Brain, Cognition and Behaviour, Radboud University Medical Center, Nijmegen, The Netherlands

[7]Division of Geriatric Epidemiology, Karolinska Institutet, Stockholm, Sweden

[8]Department of General Practice, Section of Medical Ethics, Amsterdam UMC/University of Amsterdam, Amsterdam, The Netherlands

**Correction notice** This article has been corrected since it first published online. The open access licence type has been amended.

**Acknowledgements** We thank all Finnish and Dutch nurses for their participation in the study, Suzanne Ligthart, Carin Miedema, Paulien Vermunt, Floor Rooskens (respectively, discussion moderator, assistant-moderator and assistants in transcription and coding for the Dutch meetings), Lotta Salo, Ejja Pietilä (organisation and summary notes of the Finnish meeting) and Nicola Coley (review of written English) for their assistance to the study.

**Contributors** Study design: CRLB, ER, EMvC, HS, FM and MK. Data acquisition: CRLB, MB and AR. Data analysis: CRLB, UA, MB and AR. Interpretation of results: CRLB, UA, MB, AR, FM, EMvC and AJP. Drafting of the manuscript: CRLB. Critical revision of the manuscript for important intellectual content: all authors.

**Funding** The research leading to these results has received funding from the European Union Seventh Framework Programme for the HATICE-project (FP7/2007-2013, grant agreement no 305374) and the Multimodal preventive trials for Alzheimer's Disease: towards multinational strategies—programme (MIND-AD: MIND-AD Academy of Finland (grant agreement no 291803), MIND-AD VTR, Kuopio University Hospital (grant agreement no 5772815) and MIND-AD Academy of Sweden (Swedish Research Council, 529-2014-7503).

**Competing interests** None declared.

**Patient consent for publication** Not required.

**Provenance and peer review** Not commissioned; externally peer reviewed.

**Data sharing statement** Data will not be made publicly available. For more information, please contact the corresponding author.

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
