## [Reviewer comments · BMJ Open]

ARTICLE DETAILS

TITLE (PROVISIONAL)	Integrating nurses' experiences with supporting behaviour change for cardiovascular prevention into a self-management internet-platform in Finland and the Netherlands: a qualitative study
AUTHORS	Beishuizen, Cathrien; Akenine, Ulrika; Barbera, Mariagnese; Rosenberg, Anna; Fallah Pour, Mandana; Richard, Edo; Soininen, Hilikka; Mangialasche, Francesca; Kivipelto, Miia; Pols, A.; Moll van Charante, Eric

VERSION 1 – REVIEW

REVIEWER	Howard Leventhal Department of Psychology and Institute for Health, Rutgers University, New Brunswick, NJ, United States
REVIEW RETURNED	30-Jul-2018

GENERAL COMMENTS	REVIEW: Integrating nurses' experience---- qualitative study. It's difficult to make a recommendation as to the suitability for publication of the current manuscript. My misgivings reflect issues in method and presentation. Regarding methods, repeating the mantra "grounded theory" does not compensate for the relatively low rate of participation (see 3 below) or failure to provide similar information for both sites. Most importantly, the data do not seem to reflect the nurses actual experiences with the HATICE system: their responses require them to imagine "what would happen", and that is not the same as what have you done or felt when you addressed CVD risk with specific patients using HATICE. Though focus groups are hard work, two groups, one from each country, is a very limited sample. I don't see how they could judge whether they reached "saturation"; it would have been helpful to see what else they might expect to hear from nurses in other sections of the country. It would also be extremely interesting to see how nurses in the Netherlands would respond to comments by nurses in Finland, and versa, especially where there seemed to be differences. Additional data of that type might be more interesting than variation in the age or participants, though not necessarily more interesting than comparisons among nurses from urban and rural areas, this latter picks up on the discussion. The first paragraph of the discussion summarized the questions raised in #4 below. The text surrounding the thematic content could have done a better job highlighting the issue. For example, the authors could have raised the question, "is the difference an individual or system factor" when presenting the comments from each group. The discussion could both repeat the question and ask whether particular contextual factors are responsible for the differences, as it started to do though I would have liked to read more about possible system differences in the definition of the
---

nurses role Vs that of the physician, possible differences in the interpersonal styles of the two countries, and so forth. As it is we are left with questions and forced to imagine answers.

Major Questions

1) Given that the nurses had never used HATICE, their responses are unrelated to actual patient experience. In sum, they are “imagining how they might use the platform. Their responses reflect therefore, prior experience with other internet platforms (if any) and their ability to project themselves into this as yet unknown space.

2) Why mention cognitive decline (dementia) in addition to CVD. The former is NOT discussed in both groups and, and the self-regulatory activities introduced in the intervention are focused on CVD. Not clear whether adding in dementia made a difference in discussion content, at least nothing is presented to suggest it did.

3) Seven of 32 is unimpressive; an excessive non-response rate. Do the 7 differ from the 25 non responders, and if so, how? In addition, what was the rate in Finland; did all asked participate, or is it also 6 of a larger number?

4) They could have said more about the finding that Dutch nurses attributed a more directive role to themselves and the medical practice to avoid mistakes and complications regarding the medical components of preventive care (control of hypertension, diabetes and hypercholesterolemia), whereas the Finnish nurses regarded their patients as capable of staying in charge. Is the difference a practitioner factor, or does it reflect differences in the population. For example, if the Finnish patients are more homogeneous and the Dutch far more diverse, the differences in practitioner behavior may reflect the diversity of problems faced in everyday practice. It would be useful to say more about this as it seems important.

5) The Dutch nurses seemed to differentiate the online coach from the professional engaged in person whereas the Finnish nurses seemed to see the same person in two roles. Is this difference purely accidental, someone in the group stated it as two different people, or does it reflect features of the health care systems in each country, the Dutch assigning different individuals to the two roles whereas the Finnish do not. In other words, is the difference lodged in the individuals interpersonal and nursing practices and/or styles, or are the individual nurses expressing a systemic factor? If the latter, the nurses would have relatively little difficulty behaving differently if they moved from system to system. The text is unclear however, as to whether the online individual is the same as the nurse with direct patient contact. Is there a reality here; i.e., what is actually done in hospitals where HATICE is used. In fact, has it been used, and if so, where and why wasn't a focus group conducted among nurses using HATICE.

Minor Issues

1) Second sentence of abstract – integrate these – Not clear what these refers to.

Why not rewrite as follows: -- nurses' experience with integrating behavior change guidance for cardiovascular disease into an internet-platform. –

	2) Participants and setting. The first sentence is both lengthy and not entirely clear. It is my guess that “primary care nurses experienced in “ are the same individuals as the “field experts”.
--	---

REVIEWER	ANIBAL GARCIA-SEMPERE FOUNDATION FOR BIOMEDICAL RESEARCH OF VALENCIA (FISABIO), SPAIN
REVIEW RETURNED	28-Aug-2018

GENERAL COMMENTS	This is a well-written paper reporting the results of a qualitative study aiming to elucidate what are the key elements for a selected group of nurses with experience in cardiovascular prevention from Finland and the Netherlands to enable health behavioral change towards a better self-management of cardiovascular and dementia prevention, and how to integrate nurse’s practices related to these elements into e-health patient support platforms. Trust, expectation management and appropriate planning of support were identified as key enablers of behavior change by the two groups of nurses, but their views differed with regard to the degree in which patient management should be integrated into online, telecare services. The authors conclude that characteristics of local practices should be taken into account when implementing ehealth solutions to improve chronic care. I have only very minor comments to the manuscript. Abstract/Results section: I think that the conditions for behavioral change identified are the first main result of the study and should be reported in this section of the abstract. Introduction page 3 line 27. I do not agree that little is known on the effectiveness of telehealth self-management interventions in chronic patients. There is a huge body of literature on the subject (though offering in general mixed results with regard to their effect on outcomes such a quality of life, admissions due to worsening of chronic conditions, etc). Methods: the “Design” section subheading is missing. Discussion. I am not sure the subheadings follow BMJ guidelines, please check. A Figure in the Discussion is quite uncommon, too. Discussion. The geographical factor (low population density and lower access to direct face to face healthcare in Finland than in the Netherlands) seems to me quite important with regard the perceptions of integration of nurses’ practices in ehealth solutions. We would be facing here a matter of pragmatism in front of limitations rather than preferences (writing sometimes seems to suggest the later). Discussion. The fact that clinical experience was very different between groups is reported as a limitation. Average experience of Finnish nurses was 18 years, while average experience of Dutch nurses was around 7. I think this should not only be commented as a limitation, but also as a potential explanatory factor of differences in perception and reliance on telemonitoring and ehealth remote care solutions.
--

REVIEWER	Kate Morton University of Southampton
REVIEW RETURNED	04-Oct-2018

GENERAL COMMENTS	This was an interesting paper which brings together findings from two different healthcare systems. The paper has an important message about taking the context of the healthcare system into account when developing e-health interventions. I have attached a few comments with suggestions for amendments. There are two important limitations which I think need further consideration in the discussion. Integrating nurses' experiences with supporting behaviour change for cardiovascular prevention into a selfmanagement internet-platform in Finland and the Netherlands: a qualitative study Abstract:  1. Minor editing work is needed on abstract. E.g. Double full stop after 'Netherlands'; second sentence should read 'Self-management and eHealth applications are regarded as promising strategies to support prevention' Introduction:  2. The introduction is concise and provides a good justification for the study. 3. For me, the claim that 'e-health interventions can easily support self-management' (line 24) felt a bit strong and was not backed up by evidence. The evidence is mixed in terms of effectiveness, and whether or not it is easy depends on many factors. Perhaps this could be amended? E.g. 'E-health interventions have the potential to support self-management', and include a reference to support this. Methods:  4. Participants: The manuscript doesn't say how many nurses were invited in Finland 5. It would be helpful to report the % uptake rate in each country 6. I found Box 3 very helpful as a background on the healthcare systems in each country. Perhaps you could refer to Box 3 in the Participants paragraph, where you first explain that occupational health nurses were recruited from Finland, as this is where I first wondered about the different healthcare systems in the two countries? 7. Box 3 was very useful, but I wonder if it would be even more useful displayed as a table with columns for each country, so the reader can easily understand similarities and differences. This would help contextualise some info which is less obviously relevant, e.g. I found it interesting that Finland was the first European country to define patient's right to access medical records, but I wasn't clear how this was relevant. If this was included in a table under a row heading like 'e-health culture to date' it would help the reader to see this info in context and compare it with the Netherlands. 8. From the description of Finnish healthcare, it sounds like there are nurses dealing with CV risk in public healthcare centres, so it wasn't clear to me why you decided to only target one occupational healthcare centre? Maybe this decision could be justified?
---

9. How were the participants invited? Was this done by email?
10. Typo: Data collection, page 5: "The discussion was conduction using a topic list". This should be 'conducted'.
11. It would be helpful if the authors could briefly describe how the topic guide was refined following the Dutch session, and why?
12. I found it confusing to say that 'Both moderators first discussed...' I feel that 'discussed' is the wrong word as this sounds as though the moderators told the participants about this topic. I think 'asked' would be clearer.
13. Box 1: What does 'good guidance of behaviour change' mean? Were nurses asked for examples of how they change behaviour? Or was some existing guidance discussed?
14. Some brief reflection on why focus groups were chosen as the most appropriate method to explore this topic would be useful, e.g. rather than interviews.

Results

15. The analysis is described as inductive, yet the same theme names are applied to the data in Part 1 and Part 2. For me, this raised questions about how inductive it was. If it was the case that Part 1 was analysed inductively first, and then the theme names were applied deductively to Part 2, then this should be described in the methods.
Or if the whole focus group transcripts were analysed together and the three themes derived inductively, then Parts 1 and 2 would work better for me if reported in an integrated format, under each theme heading. I think it would work well to start each theme description with the preconditions and then the implications for the online tool.
Otherwise it seems to artificially imply that the same themes emerged from both sections of the data.
16. Interpretation of quotes: Quote 1 – Dutch nurse 1. This quote seems to suggest that the nurse believes that trust comes from a long-standing relationship, but this is not discussed in the text.
Quote 2 – Finnish nurse 1. For me, this quote didn't say anything about trust, and didn't support the comment in the text that as the nurse doesn't say she uses these skills in order to increase trust. It sounds like the nurse is talking more about what kind of support will be most helpful to patients?
17. The discussion about different modes of communication does not seem to fit under this theme of 'trust'. The quotes around this seem to relate more to the convenience of email contact, not trust. Perhaps some other quotes might demonstrate this better, and the text describing this finding could be more clearly related to the theme?
18. The quote from Dutch nurse 3 does not seem to correspond to the text. The text suggests nurses had to reshape goals because people were aiming too high, but the quote is about people not wanting to go to the gym at all and how they get around this by suggesting small steps at first.
19. 'the Dutch nurses attributed a more directive role to themselves and the medical practice to avoid mistakes and complications, whereas the Finnish nurses regarded their patients as capable of staying in charge'. This is a really interesting observation, I wondered if it would be possible to include a quote or two to demonstrate the different

perceptions about self-management. Were there any exceptions (deviant cases) who didn't fit this pattern?

20. The final quote does not correspond with the text, which is about the platform being suitable for lifestyle change only, not medical self-management.

21. Typo: An apostrophe is needed after the 's' of nurses in the title: 'Nurses experiences and practices with supporting the process of behaviour change for cardiovascular prevention

22. Typo: Line 39, page 6 – I think it should say 'establishing a relationship of trust'

23. Typo: Line 40, page 6 – "the basis of behaviour change support lied in establishing a relationship of trust'. I think this should be 'lay'.

Discussion

24. Para 1: I feel that it is misleading to state that 'they regarded these preconditions equally important', as the data are qualitative so it's not possible to say whether the three preconditions were regarded as equally important or not. To me, this language implies quantitative data. Perhaps a more qual phrasing could be used, e.g. 'all three preconditions were regarded as important by participants'.

25. The discussion is interesting and brings together the findings nicely. It helped me understand why the findings about email vs face to face contact were included under the 'trust' theme, but I think this needs to be brought out more clearly in the findings.

26. I would suggest a new para at the point of 'Our results concerning dementia prevention' to make it easier for the reader

27. Line 56, I would suggest using a different word in place of 'enlarge' for describing motivation. Perhaps 'increase'?

28. The first sentence under 'comparison with existing literature' felt very broad and non-specific, perhaps you could explain in a bit more detail how the findings were consistent with the Dutch studies you have cited (refs 27-31)?

29. You mention in the strengths and limitations that the two groups were not comparable in age, and this might have influenced the findings. I would suggest why you think age might be important, or not mention this at all.

30. I wondered about the difference in the timepoints of your data collection, and whether that needed to be considered when comparing the groups' beliefs about online support vs face to face support. The Dutch group were less receptive to the idea of online support, but the focus group was also conducted 2 years earlier. Perhaps this difference in timepoints should be mentioned as a potential limitation, as the developments in e-health in this time might have encouraged the Finnish group to feel more accepting of this?

31. It might also be worth reflecting on how your interview schedule could have influenced your findings. Some of the questions I found quite leading, for example:

"Which factors could contribute to a good relationship with your patient?" This might explain why you had a theme emerging on trust.

"Most people are not yet aware of the association between CV risk and dementia, but do seem to be very afraid of dementia. Do you think that more awareness would enhance compliance/adherence to lifestyle change?" This seems to lead people towards agreeing

	that awareness of dementia would increase adherence, which was one of your findings. Appendices: 32. I like that you used the COREQ checklist. Would it be possible to add details of items 1-4 to the methods section, so the reader understands these details at the time?
--	---

VERSION 1 – AUTHOR RESPONSE

Responses to reviewer #1:

Reviewer: 1

Reviewer Name: Howard Leventhal

Institution and Country: Department of Psychology and Institute for Health, Rutgers University, New Brunswick, NJ, United States

Please state any competing interests or state 'None declared': None

Please leave your comments for the authors below

REVIEW: Integrating nurses' experience---- qualitative study.

General comment 1: It's difficult to make a recommendation as to the suitability for publication of the current manuscript. My misgivings reflect issues in method and presentation. Regarding methods, repeating the mantra "grounded theory" does not compensate for the relatively low rate of participation (see 3 below) or failure to provide similar information for both sites. Most importantly, the data do not seem to reflect the nurses actual experiences with the HATICE system: their responses require them to imagine "what would happen", and that is not the same as what have you done or felt when you addressed CVD risk with specific patients using HATICE. Though focus groups are hard work, two groups, one from each country, is a very limited sample. I don't see how they could judge whether they reached "saturation"; it would have been helpful to see what else they might expect to hear from nurses in other sections of the country. It would also be extremely interesting to see how nurses in the Netherlands would respond to comments by nurses in Finland, and versa, especially where there seemed to be differences. Additional data of that type might be more interesting than variation in the age or participants, though not necessarily more interesting than comparisons among nurses from urban and rural areas, this latter picks up on the discussion.

Response: We thank the reviewer for his thorough evaluation of our manuscript and efforts taken to explain his misgivings. We understand his concerns regarding the methodological limitations of our research. We hope that our elaboration on his comments will lead to more clarity and easier interpretation for the reader regarding the methodological limitations that we cannot overcome, but also to a better understanding and justification of the methodological approach we chose in this research project.

The first paragraph of the discussion summarized the questions raised in #4 below. The text surrounding the thematic content could have done a better job highlighting the issue. For example, the authors could have raised the question, "is the difference an individual or system factor" when presenting the comments from each group. The discussion could both repeat the question and ask whether particular contextual factors are responsible for the differences, as it started to do though I would have liked to read more about possible system differences in the definition of the nurses role Vs that of the physician, possible differences in the interpersonal

styles of the two countries, and so forth. As it is we are left with questions and forced to imaging answers.

Response: We thank the reviewer for his suggestions to clarify the discussion. We have adapted the discussion to better answer the question the reviewer raises (“is the difference an individual or system factor”). In short, we think that most of our findings direct towards system factors, being the differences between health care cultures (organisation of care and level of patient empowerment), and geographical differences. However, due to some sample-differences (level of clinical experience and patient population age) we cannot exclude that individual factors played a role. Please, see the elaboration of these points in the responses to major questions 4 and 5.

Major Questions:

1) Given that the nurses had never used HATICE, their responses are unrelated to actual patient experience. In sum, they are “imagining how they might use the platform. Their responses reflect therefore, prior experience with other internet platforms (if any) and their ability to project themselves into this as yet unknown space.

Response: We understand the reviewer’s doubts regarding our chosen approach to ask nurses experienced in ‘traditional’ face-to-face cardiovascular preventive care to give us recommendations on - at the time- ‘fictional’ online cardiovascular preventive care. What would they know about online support if they have not used it before? However, we think this approach is valid and highly appropriate to meet our research objectives. Here we try to justify this. Our research aim was to gain recommendations for online support of health behaviour change. Face-to-face support can still be regarded the ‘gold standard’ in health behaviour change programs, because effectiveness increases with program-intensity and face-to-face programs have a higher intensity than online programs (see for example: Patnode et al. Behavioral Counseling to Promote a Healthful Diet and Physical Activity for Cardiovascular Disease Prevention in Adults Without Known Cardiovascular Disease Risk Factors: Updated Systematic Review for the U.S. Preventive Services Task Force. 2017). Seen in that light, it is highly informative to ask experts in face-to-face support about their best practices for health behaviour change and how you could preserve these practices in an online support environment. This is also a common research approach in Grounded Theory (trying to understand a ‘practice’ or ‘process’ by consulting people that are experts in this process). We clarified our rationale in the introduction by adding:

“Although researchers and policymakers have high expectations of eHealth and self-management, the more intensive face-to-face interventions still achieve better results than eHealth applications²⁰. To learn how self-management and behaviour change are best stimulated and maintained online, we consulted ‘experts in the field’: nurses experienced in health behaviour change in the context of CV prevention.” (Introduction, p4, lines 15-19)

2) Why mention cognitive decline (dementia) in addition to CVD. The former is NOT discussed in both groups and, and the self-regulatory activities introduced in the intervention are focused on CVD. Not clear whether adding in dementia made a difference in discussion content, at least nothing is presented to suggest it did.

Response: We understand that the reviewer questions why we mention prevention of dementia in addition to CVD. We agree that it is a serious limitation that dementia prevention was only discussed with the Finnish nurses. During the writing of the results we have considered to leave out the findings on dementia prevention. However, we think these findings do make a difference to the discussion content, because, although very preliminary, they provide novel opportunities to frame health behaviour change for both prevention of dementia and CVD. We have adapted our discussion of these findings by better emphasizing the preliminary character of the findings:

“Our results concerning dementia prevention are very preliminary but of special interest.”(Discussion, p15, line 1)

“A further limitation is that cognitive health was only discussed with the Finnish nurses.”
(Discussion, p15, lines 27-28)

“Including the maintenance of cognitive health as a goal of cardiovascular prevention can provide novel opportunities to frame health behaviour change for both prevention of dementia and CVD and might augment people’s motivation for prevention, but this suggestion should be studied further.”(Discussion, p16, lines 17-20)

3) Seven of 32 is unimpressive; an excessive non-response rate. Do the 7 differ from the 25 non responders, and if so, how? In addition, what was the rate in Finland; did all asked participate, or is it also 6 of a larger number?

Response: We agree with the reviewer that the Dutch non-response rate is rather high. The 25 practice nurses that chose not to participate did not differ from the responders, they were also female practice nurses working in general practices in the same region and they had the same amount of experience with cardiovascular preventive care. Their unanimous reason not to participate was lack of time. For the Finnish data, information on non-response was retrieved. In Finland, 14 nurses worked in the semi-private health care centre. Of those 6 (43%) consented to participate. Reason for nonparticipation was also lack of time. We clarified this information in the methods:

“Fourteen nurses working in a semi-private healthcare centre in Kuopio (Eastern Finland) were invited by email and telephone and six female nurses (43%) consented to participate.” (Methods, p5, lines 11-13)

“A group of 32 nurses experienced in CV preventive care working in general practices in two urban areas in the centre of the Netherlands was invited by email and telephone. Seven female nurses (22%) consented to participate. The unanimous reason for non-participation by Finnish and Dutch nurses was lack of time.” (Methods, p5, lines 15-18)

We emphasized this limitation in the discussion by adding:

“Our research also has some limitations that may have influenced our findings. Information on non-participation was limited.” (Discussion, p15, lines 19)

4) They could have said more about the finding that Dutch nurses attributed a more directive role to themselves and the medical practice to avoid mistakes and complications regarding the medical components of preventive care (control of hypertension, diabetes and hypercholesterolemia), whereas the Finnish nurses regarded their patients as capable of staying in charge. Is the difference a practitioner factor, or does it reflect differences in the population. For example, if the Finnish patients are more homogeneous and the Dutch far more diverse, the differences in practitioner behavior may reflect the diversity of problems faced in everyday practice. It would be useful to say more about this as it seems important.

Response: We appreciate that the reviewer emphasizes this finding, as we also regard it important. We think this difference derives from 3 factors: the practitioner, the patient and the underlying health care culture. To start with the latter, as we describe in Box 3 (Finnish and Dutch primary health care systems), and in the discussion (p14), patient- empowerment and patient autonomy are more developed in Finnish health care culture than in the Dutch one. This influences how the nurses support their patients, being more paternalistic in the Netherlands versus more patient-centered in Finland. In the discussion, we hypothesise that geographical

differences also play a role. If the nearest primary care clinic is 40 kilometres away, you will only go there if you feel you cannot solve problems yourself any longer. In the Netherlands the threshold to ask for the doctor's advice is much lower with patients all living in close vicinity of the GP surgery. Last, it seems plausible that our findings are also influenced by age, since the Finnish patients were on average younger than the Dutch patients, and the nurses might allocate more health autonomy and eHealth eagerness to their younger patients than to their older patients. So possibly the contrast between the attitude of the Finnish and Dutch nurses has been amplified in our research setting.

We have completely adapted the first paragraph of the discussion to further elaborate on our finding of nurses' different attitudes towards patient autonomy.

5) The Dutch nurses seemed to differentiate the online coach from the professional engaged in person whereas the Finnish nurses seemed to see the same person in two roles. Is this difference purely accidental, someone in the group stated it as two different people, or does it reflect features of the health care systems in each country, the Dutch assigning different individuals to the two roles whereas the Finnish do not. In other words, is the difference lodged in the individuals interpersonal and nursing practices and/or styles, or are the individual nurses expressing a systemic factor? If the latter, the nurses would have relatively little difficulty behaving differently if they moved from system to system. The text is unclear however, as to whether the online individual is the same as the nurse with direct patient contact. Is there a reality here; i.e., what is actually done in hospitals where HATICE is used. In fact, has it been used, and if so, where and why wasn't a focus group conducted among nurses using HATICE.

Response: At the time the focus groups were performed, the HATICE platform was still in its building phase and had not been used yet. Currently, the HATICE platform has been tested in a large RCT, with results in the process of publication.

The reviewer understood correctly that, when we asked both Dutch and Finnish nurses whether they felt comfortable with exercising their role purely through internet, the Dutch nurses felt hesitant and wanted to divide their nursing tasks into low risk support, that could be allocated to an online coach, and high risk support, that should be preserved for the GP surgery. In contrast, the Finnish nurses easily envisioned themselves performing all nursing tasks online. These differences mainly originate from the different attitude towards patient autonomy the nurses have, which is lodged in the local health care practices, as has been elaborated in the response to #4. The views of the nurses are in line with 'common' styles of caring in the Netherlands and Finland. For the Netherlands, we know that these more paternalistic views are shared by general practitioners too, (see discussion paragraph Comparison with existing literature). As the second reviewer suggests, some individual characteristics may also be of influence, such as extent of clinical experience and average age of the patient populations. These nuances are further clarified in the discussion. To further study our findings, it would be very interesting, as the reviewer suggests, to ask the same questions to nurses who have actual experience with online support systems. In an evaluation of HATICE, we have done so with the health coaches. However, these findings are not ready for publication yet.

Minor Issues

1) Second sentence of abstract – integrate these – Not clear what these refers to.

Why not rewrite as follows: -- nurses' experience with integrating behavior change guidance for cardiovascular disease into an internet-platform. –

Response: We thank the reviewer for the suggestion and have adapted as follows:

The aim of this study was to explore primary care nurses' best practices with behaviour change guidance for cardiovascular (CV) prevention in order to learn how to optimally integrate these into

a coach-supported internet-platform with coaching for cardiovascular CV self-management.

2) Participants and setting. The first sentence is both lengthy and not entirely clear. It is my guess that “primary care nurses experienced in “ are the same individuals as the “field experts”.

Response: To clarify and shorten the sentence, we adapted as follows:

“For sampling, we followed the grounded theory methodology of studying a health care practice by consulting field experts ²⁵. In this light, Finnish and Dutch primary care nurses experienced in CV preventive care were most eligible for this study and selective purposive samples were obtained.”(Methods, p5, lines 6-9)

Reviewer: 2

Reviewer Name: ANIBAL GARCIA-SEMPERE

Institution and Country: FOUNDATION FOR BIOMEDICAL RESEARCH OF VALENCIA (FISABIO), SPAIN

Please state any competing interests or state ‘None declared’: None declared

Please leave your comments for the authors below

This is a well-written paper reporting the results of a qualitative study aiming to elucidate what are the key elements for a selected group of nurses with experience in cardiovascular prevention from Finland and the Netherlands to enable health behavioral change towards a better self-management of cardiovascular and dementia prevention, and how to integrate nurse’s practices related to these elements into e-health patient support platforms. Trust, expectation management and appropriate planning of support were identified as key enablers of behavior change by the two groups of nurses, but their views differed with regard to the degree in which patient management should be integrated into online, telecare services. The authors conclude that characteristics of local practices should be taken into account when implementing ehealth solutions to improve chronic care.

I have only very minor comments to the manuscript.

1) Abstract/Results section: I think that the conditions for behavioral change identified are the first main result of the study and should be reported in this section of the abstract.

Response: We agree with the reviewer and added to the results section of the Abstract:

“Similar best practices were found and comprised of: (1) establishing a relationship of trust, (2) managing awareness and expectations and (3) appropriate timing and monitoring of the process of behaviour change.”

2) Introduction page 3 line 27. I do not agree that little is known on the effectiveness of telehealth self-management interventions in chronic patients. There is a huge body of literature on the subject (though offering in general mixed results with regard to their effect on outcomes such a quality of life, admissions due to worsening of chronic conditions, etc).

Response: We agree with the reviewer that quite a body of evidence exists on the effectiveness of eHealth self-management interventions for chronic conditions and even components of CV prevention on various intermediate outcomes. However, in the paper we aim to focus on the process of supporting health behaviour change, and, although this is also increasingly being studied in online applications, still little is known on how to do this optimally (see for example, Shingleton et al. Technology-delivered adaptations of motivational interviewing for health-related

behaviors: A systematic review of the current research. Pat Edu & Counselling 2016).
We adjusted the phrasing to emphasise our focus and methodological approach:

“Although researchers and policymakers have high expectations of eHealth and self-management, the more intensive face-to-face interventions still achieve better results than eHealth applications²⁰. To learn how self-management and behaviour change are best stimulated and maintained online, we consulted ‘experts in the field’: nurses experienced in health behaviour change in the context of CV prevention.” (Introduction, p4, lines 15-19)

3) Methods: the “Design” section subheading is missing.

Response: the “Design”- subheading was added to the first paragraph of the methods.

4) Discussion. I am not sure the subheadings follow BMJ guidelines, please check. A Figure in the Discussion is quite uncommon, too.

Response: To better follow the BMJ-open guidelines, we changed the order of 2 paragraphs, to have ‘Principal findings’, followed by “Strengths and limitations”, followed by “Comparison with existing literature”.

We agree with the reviewer that it is quite uncommon to have a figure in the Discussion. However, in qualitative research we believe it to be appropriate, as the figure depicts a synthesis and interpretation of the results. Since the other reviewers did not have comments on this point, we propose to keep the figure in the Discussion.

5) Discussion. The geographical factor (low population density and lower access to direct face to face healthcare in Finland than in the Netherlands) seems to me quite important with regard the perceptions of integration of nurses’ practices in ehealth solutions. We would be facing here a matter of pragmatism in front of limitations rather than preferences (writing sometimes seems to suggest the later).

Response: We agree with the reviewer that the geographical factor is rather important as an explanatory factor of our findings. However, we do believe that this factor includes more than solely pragmatism. Pragmatism can be described as an organisational factor. We believe that the geographical factor also has influenced the level of patient empowerment in both countries. We tried to support this in Box 3. We also completely adapted the first paragraph of the discussion to better put the geographical factor in context.

6) Discussion. The fact that clinical experience was very different between groups is reported as a limitation. Average experience of Finnish nurses was 18 years, while average experience of Dutch nurses was around 7. I think this should not only be commented as a limitation, but also as a potential explanatory factor of differences in perception and reliance on telemonitoring and ehealth remote care solutions.

Response: The reviewer raises an interesting point but we find it difficult to support the suggestions with our data. Although the clinical experience with CV prevention of the Dutch nurses was lower, general clinical experience as practice nurse was comparable between the 2 groups.

Reviewer: 3

Reviewer Name: Kate Morton

Institution and Country: University of Southampton

Please state any competing interests or state 'None declared': None declared

Please leave your comments for the authors below

This was an interesting paper which brings together findings from two different healthcare systems. The paper has an important message about taking the context of the healthcare system into account when developing e-health interventions.

I have attached a few comments with suggestions for amendments. There are two important limitations which I think need further consideration in the discussion.

Abstract:

1. Minor editing work is needed on abstract. E.g. Double full stop after 'Netherlands'; second sentence should read 'Self-management and eHealth applications are regarded as promising strategies to support prevention'.

Response: We have made these corrections in the abstract (abstract, p2, lines 5 and 9)

Introduction:

2. The introduction is concise and provides a good justification for the study.

3. For me, the claim that 'eHealth interventions can easily support self-management' (line 24) felt a bit strong and was not backed up by evidence.

The evidence is mixed in terms of effectiveness, and whether or not it is easy depends on many factors. Perhaps this could be amended. E.g. 'EHealth interventions have the potential to support self-management', and include a reference to support this.

Response: We understand the reviewer's thoughts on our statement. What we intended to say is that eHealth applications seem to be suitable devices to support self-management because of technical possibilities to facilitate health education, interactive use and health monitoring. We clarified this by moderating the statement and including a justification. We think that the two references back up our statement sufficiently, since these are two reviews about the potential and possibilities of eHealth to deliver health interventions.

"eHealth applications are attractive because of their wide reach and have the potential to support self-management because of their suitability for health education, interactivity and monitoring"^{18 19} (introduction, p4, lines 13-15)

Methods:

4: Participants: the manuscript doesn't say how many nurses were invited in Finland.

Response: This information was retrieved and included as follows:

"Fourteen nurses working in a semi-private healthcare centre in Kuopio (Eastern Finland) were invited by email and telephone and six female nurses (43%) consented to participate." (Methods, p5, lines 11-13)

5: It would be helpful to report the % uptake rate in each country.

Response: We added this information for each country:

"Fourteen nurses working in a semi-private healthcare centre in Kuopio (Eastern Finland) were invited by email and telephone and six female nurses (43%) consented to participate." (Methods, p5, lines 11-13)

"A group of 32 nurses experienced in CV preventive care working in general practices in two urban areas in the centre of the Netherlands was invited. Seven female nurses (22%) consented

to participate. The unanimous reason for non-participation was lack of time.”(Methods, p5, lines 15-17)

6: I found Box 3 very helpful as a background on the healthcare systems in each country. Perhaps you could refer to Box 3 in the Participants paragraph, where you first explain that occupational health nurses were recruited from Finland, as this is where I first wondered about the different healthcare systems in the two countries?

Response: We agree with the reviewer and moved the introduction of Box 3 to the Participants paragraph:

“In Finland, we recruited occupational healthcare nurses because of their important role in preventive CV care (please see **Box 3** for a description of Finnish and Dutch primary health care systems and the position of Finnish occupational health).” (Methods, p5, lines 8-11)

7: Box 3 was very useful, but I wonder if it would be even more useful displayed as a table with columns for each country, so the reader can easily understand similarities and differences. This would help contextualize some info which is less obviously relevant, e.g. I found it interesting that Finland was the first European country to define patient’s right to access medical records, but I wasn’t clear how this was relevant. If this was included in a table under a row heading like ‘e-health culture to date’ it would help the reader to see this info in context and compare it with the Netherlands.

Response: We thank the reviewer for this suggestion and executed it as we agree this would very much help the reader to contextualize the information presented. (Box 3, page 21).

8: From the description of Finnish health care, it sounds like there are nurses dealing with CV risk in public health care centres, so it wasn’t clear to me why you decided to only target one occupational health care centre. Maybe this decision could be justified?

Response: As described in Box 3, primary care nurses and occupational health care nurses have very similar roles and tasks, especially when it comes to CV prevention. In that respect, and because occupational health care facilities have very similar position next to public primary care, they were just as eligible for our study as public primary care nurses. Since their tasks in CV prevention may be even bigger than those of public primary care nurses, we choose them. We targeted one centre, because this yielded sufficient nurses willing to participate.

We clarified this as followed:

“In Finland, occupational healthcare nurses were recruited because of their important role in preventive CV care (please see **Box 3** for a description of Finnish and Dutch primary health care systems and the position of Finnish occupational health).” (Methods, p5, lines8-11)

9: How were the participants invited? Was this done by email?

Response: we invited the nurses by email and telephone. This information was added to the methods.

10: Typo: Data collection, page 5: “The discussion was conduction using a topic list”. This should be ‘conducted’.

Response: this typo was corrected. (p6, line 19)

11: It would be helpful if the authors could briefly describe how the topic guide was refined following the Dutch session, and why?

Response: After the Dutch session, the Dutch team started with first analyses of the Dutch transcripts. Based on those, we had an idea on which topics were most relevant and merited more in depth exploration. These topics were the ones that the Dutch nurses expressed the strongest ideas making it most interesting to compare with the experiences, ideas and attitudes of the Finnish nurses. These topics included: the nurse-patient relationship, attitude towards eHealth applications, shaping optimal online support and the role of the nurse in this, responsibilities of the patient versus those of the nurse and what online support should include. Second: we put more emphasis on the discussion of the topic 'dementia prevention', because this was missed in the Dutch focus group. We added this to the methods as follows:

"After the Dutch session, the topic list was refined for the Finnish focus group, to further explore the following topics: the nurse-patient relationship, attitude towards eHealth applications, shaping optimal online support, the role of the nurse in this versus the role of the patient and dementia prevention"(methods, p 6, lines 16-19)

12: I found it confusing to say that 'Both moderators first discussed...' I feel that 'discussed' is the wrong word as this sounds as though the moderators told the participants about this topic. I think 'asked' would be clearer.

Response: We changed this to 'asked about' (Methods, p6, lines 19-22)

13: Box 1: What does 'good guidance of behavior change' mean? Were the nurses asked for examples of how they change behavior? Or was some existing guidance discussed?

Response: We did not discuss some existing guidance or definition but asked the nurses what their experience was with CV prevention and lifestyle and health behaviour change, we asked what worked and what didn't and what they considered good guidance. So we explored the concept inductively.

14: Some brief reflection on why focus groups were chosen as the most appropriate method to explore this topic would be useful, e.g. rather than interviews.

Response: We regarded focus groups the most appropriate method to answer our research aims, because we wanted to explore the experiences and ideas of the nurses as a group, when they discuss the topics with each other and develop their ideas during the discussion. This was especially appropriate for the second part of the focus groups, where we wanted them to develop their recommendations on CV prevention through eHealth, based on their experiences of best practices in health behavior change. We also hypothesized that the nurses would have different attitudes towards eHealth, and in a focus group discussion they could respond to each other's opinions. We added a brief reflection to the methods:

"We regarded focus groups the most appropriate method to answer our research aims, because it enabled us to explore the experiences and attitudes of the nurses most completely, as the nurses could directly respond to each other's opinions and develop their ideas through the discussion. "
(methods, p6, lines 14-16)

Results

15: The analysis is described as inductive, yet the same theme names are applied to the data in Part 1 and Part 2. For me, this raised questions about how inductive it was. If it was the case that Part 1 was analysed inductively first, and then the theme names were applied deductively to Part 2, then this should be described in the methods.

Or if the whole focus groups transcripts were analysed together and the three themes derived inductively, then Parts 1 and 2 would work better for me if reported in an integrated format, under

each theme heading. I think it would work well to start each theme description with the preconditions and then the implications for the online tool. Otherwise it seems to artificially imply that the same themes emerged from both sections of the data.

Response: The way we set up the focus groups and used the topic lists we used an entirely inductive approach. We understand the reviewer's hesitations on this with our presentation of the results. We will try to explain this. In the second part of the focus group, we presented screenshots of the HATICE internet-platform and asked the nurses to imagine how online support for CV prevention should be shaped. We then asked the open question how they would optimally support their patients online. As the reviewer can see in the topic guide, we used open questions to concretize this, such as: what information would you need about your patients to be able to support them, what modes of communication, how often, what kind of role would you have, how much responsibility. Through these questions they expressed their conviction that the same preconditions would also apply for optimal online support. We therefore think these results came from an inductive inquiry. It was during the iterations with the research team that we came to structure the findings as presented in the paper. We tried to clarify this process by adding to the introduction of Part 2:

"After having identified the preconditions for optimal behaviour change support and the skills nurses use in their current practices, we demonstrated the latest version of the HATICE internet-platform²⁶ and discussed how optimal online-support for CV prevention should be shaped and how they imagined providing online support." (Results, p11, lines 16-19)

16: Interpretation of quotes: Quote 1 - Dutch nurse 1. This quote seems to suggest that the nurse believes that trust comes from a long-standing relationship, but this is not discussed in the text. Quote 2 –Finnish nurse 1. For me, this quote didn't say anything about trust, and didn't support the comment in the text that as the nurse doesn't say she uses these skills in order to increase trust. It sounds like the nurse is talking more about what kind of support will be most helpful to patients?

Response:

To quote 1: we added in the text a reference to the time-element that the nurses mentioned. (p 8, line 12)

To quote 2: The reviewer is right that this quote is not about trust, it is intended as an illustration of the skill 'personalising and tailoring support'. We clarified this in the text. (p 8, line 18-19)

17. The discussion about different modes of communication does not seem to fit under this theme of 'trust'. The quotes around this seem to relate more to the convenience of email contact, not trust. Perhaps some other quotes might demonstrate this better, and the text describing this finding could be more clearly related to the theme?

Response: To us this is very essentially linked to the theme 'trust', since communication and contact moments shape the nurse-patient relationship. We tried to clarify this by adding: "emphasized the importance of face-to-face contact and in-person continuity to establish a good relationship" (p8, lines 24-25). It is true that the Finnish quotes seem also related to convenience though. We added another quote that demonstrates the importance the Finnish nurses attributed to a first face-to-face contact.

"And here [refers to the HATICE platform] the initial contact and information session at the beginning is very important because I guess a sort of a relationship needs to be established here as well. In the same way. There are still people behind this platform." (Finnish nurse 1) (p9, lines 1-3)

18: The quote from Dutch nurse 3 does not seem to correspond to the text. The text suggests nurses had to reshape goals because people were aiming too high, but the quote is about people not wanting to go to the gym at all and how they get around this by suggesting small steps at first.

Response: The quote was intended to elaborate on the text and provide an example of the skills the nurses use to help their patients making the goals achievable and into a positive, stimulating experience. We adjusted the text to better link with the quote:

“Often, once people were motivated to change their health behaviours, they also tended to set unrealistic goals, which the nurses then needed to bring back to realistic proportions:”(p9, lines 24-25)

19: ‘the Dutch attributed a more directive role to themselves and the medical practice to avoid mistakes and complications, whereas the Finnish nurses regarded their patients as capable of staying in charge’. This is a really interesting observation, I wondered if it would be possible to include a quote or two to demonstrate the different perceptions about self-management. Were there any exceptions (deviant cases) who didn’t fit this pattern?

Response: The nurses did not give direct clear-cut definitions of self-management. Their perceptions got shape during the discussion, sharing and comparing experiences and situations. It became even more clear, especially for the Dutch nurses, when we asked them to envision the online-setting. During the process of the discussion, first there was ambivalence, and different opinions and ideas were expressed, but towards the end of the discussion, the nurses had developed common recommendations and their attitudes became clear. We tried to better illustrate this process in the results section, by added one quote of the Finnish nurses and one of the Dutch.

Finnish quote:

“It is also one of the nurse’s responsibilities to be a contact person, support and a sort of mentor and also to refer the patient to a doctor if the nurse notices that something is going wrong. “(Finnish nurse 4) (Results, Part 1, p11, lines 13-14)

Dutch quote:

“I tend to think: if it is self-management, you shouldn't want to get yourself involved in that [medication use], you should leave that with the GP. On the other hand, if someone’s blood pressure is constantly rising, then you do want to know which medication someone is taking, to get the complete picture. Because then you check whether there might be a problem in medication-use.” (Dutch nurse 3) (Results, Part 2, p13, lines 9-10)

20. The final quote does not correspond with the text, which is about the platform being suitable for lifestyle change only, not medical self-management.

Response: We moved the quote, and slightly altered the text, so it better corresponds with the text. (p13, lines 1-5)

21: Typo: An apostrophe is needed after the ‘s’s of nurses in the title: ‘Nurses experiences and practices with supporting the process of behavior change for cardiovascular prevention’

Response: This was corrected (p8, line 1)

22: Typo: Line 39, page 6 – I think it should say ‘establishing a relationship of trust’.

Response: this was corrected (p8, line 6)

23: Typo: Line 40, page 6 – ‘the basis of behavior change support lied in establishing a relationship of trust.’ I think it should be ‘lay’.

Response: this was corrected (p8 line 11)

Discussion

24: Para 1: I feel that it is misleading to state that ‘they regarded these preconditions equally important’, as the data are qualitative so it’s not possible to say whether the three preconditions were regarded as equally important or not. To me, this language implies quantitative data. Perhaps a more qual phrasing could be used, e.g. ‘all three preconditions were regarded as important by participants’.

Response: We thank the reviewer for this critical note. Our intent was to state that they regarded these 3 preconditions just as important in the setting of face-to-face support as in the setting of online support. So the ‘equally’ was directed on the setting and not on the comparison of the 3 preconditions. We clarified this by adapted the phrasing:

“These preconditions were regarded as important also to provide optimal online support.”
(discussion, p 14, lines 8-9)

25: The discussion is interesting and brings together the findings nicely. It helped me understand why the findings about email vs face to face contact were included under the ‘trust’ theme, but I think this needs to be brought out more clearly in the findings.

Response: We attempted to clarify this by better linking the mode of communication to the ‘trust’ theme, already in ‘Part 1 of the results, as the mode of communication was an important aspect of the nurses’ skills of establishing a relationship of trust:

“Interestingly, when further exploring these skills, the nurses expressed different preferences regarding the ideal mode of communication. The Dutch nurses emphasized the importance of face-to-face contact and in-person continuity to establish a good relationship. For Finnish nurses an initial face-to-face contact only seemed sufficient to establish a working relation:“ (Results, p. 8, lines 23-26)

26: I would suggest a new para at the point of ‘Our results concerning dementia prevention’ to make it easier for the reader.

Response: we followed this suggestion.

27: Line 56, I would suggest using a different word in place of ‘enlarge’ for describing motivation. Perhaps ‘increase’?

Response: this suggestion was followed. (Discussion, p15, line 3)

28: The first sentence under ‘comparison with existing literature’ felt very broad and non-specific, perhaps you could explain in a bit more detail how the findings were consistent with the Dutch studies you have cited (refs 27-31)?

Response: We have specified this as follows:

“The importance of a relationship of trust, clarifying patients expectations and providing personally tailored support where also main themes in other European qualitative studies on cardiovascular preventive care with nurses or patients²⁸⁻³²” (Discussion, p15-16, lines 36-1)

29: You mention in the strengths and limitations that the two groups were not comparable in age, and this might have influenced the findings. I would suggest why you think age might be important, or not mention it at all.

Response: Here, we are referring to the mean age of the patient populations of the nurses. Reason that we think this might have influenced our findings is that a younger population may be more keen on its health autonomy than an older population, and nurses may also be more positive about using eHealth with younger than with older patients. Since the other two reviewers also stressed these points, we added these considerations to the discussion:

“Last, the differences between the patient populations’ age may have also influenced our findings, since the nurses might see more potential for eHealth applications with younger patients and regard them as more autonomous in their health behaviours.” (p14, lines 34 -37)

30: I wondered about the difference in the timepoints of your data collection, and whether that needed to be considered when comparing the groups’ beliefs about online support vs face to face support. The Dutch group were less receptive to the idea of online support, but the focus group was also conducted 2 years earlier. Perhaps this difference in timepoints should be mentioned as a potential limitation, as the developments in e-health in this time might have encouraged the Finnish group to feel more accepting of this?

Response: We agree with the reviewer that a gap of 2 years between the focus groups is rather large. However, we do not think it has influenced our findings in the way the reviewer suggests, because eHealth was also already a ‘buzz-word’ in Dutch health care in 2013, and no structural eHealth implementations took place in the work domain of the Finnish nurses in between 2013 and 2015. We therefore chose not to add this consideration to the discussion.

31: It might also be worth reflecting on how your interview schedule could have influenced your findings. Some of the questions I found quite leading, for example:

“Which factors could contribute to a good relationship with your patient?” This might explain why you had a theme emerging on trust.

“Most people are not yet very aware of the association between CV risk and dementia, but do seem to be very afraid of dementia. Do you think that more awareness would enhance compliance/adherence to lifestyle change?” This seems to lead people towards agreeing that awareness of dementia would increase adherence, which was one of your findings.

Response: We understand that the phrasing of these questions in the topic guide raises the impression that they were leading. However, the topic guide was really used as a ‘guide’ by the moderators and they were instructed to ask ‘open’ questions and to let the discussion between the nurses develop freely. In fact, in the first Dutch focus group, the opening question was asked as followed: “We like to talk about your experiences with cardiovascular prevention. What works. Lifestyle change is difficult. What are your experiences of what works and what doesn’t” And then the first response of one of the nurses was that you need to have a personal attachment with your patients. And then others added that it is a relationship of trust that is needed. So we did not really asked the ‘leading’ question, the theme really came from the nurses. In the Finnish focus group, the moderator also first asked openly about the nurses’ experiences with cardiovascular prevention and lifestyle change. We added to the methods section the following:

“The discussion was conducted using a topic list as a flexible guide.” (p6, lines 18-19)

With regard to the part on dementia prevention, questions were also asked in a more open manner than the topic list suggests. The moderator first asks what the nurses’ attitude and ideas were towards dementia prevention. This already yielded a lot of discussion from the Finnish nurses. The nurses spontaneously mentioned that dementia was stigmatized and many people had a fear of dementia. Then later, the link with CVD prevention and possibilities to enhance

motivation was discussed. This may have been a bit leading. We added in the limitations section that the findings on dementia are preliminary and need further research, since they were only discussed with the Finnish nurses:

“A further limitation is that cognitive health was only discussed with the Finnish nurses. This issue should be elaborated further in future studies.” (p15, lines 27-29)

Appendices

32: I like that you used the COREQ checklist. Would it be possible to add details of items 1-4 to the methods section, so the reader understands these details at the time?

Response: We understands the reviewers point that it would be preferable to add these details in the main text, however, due to word counts constrains, we cannot follow the suggestion.

FORMATTING AMENDMENTS (if any)

Required amendments will be listed here; please include these changes in your revised version:

- Kindly re-upload each figure under 'Image' file designation with at least 300 dpi resolution and at least 90mm x 90mm of width.

Response: we have re-uploaded the figure with the right resolution and width.

- We have implemented an additional requirement to all articles to include 'Patient and Public Involvement' statement within the main text of your main document. Please refer below for more information regarding this new instruction:

Authors must include a statement in the methods section of the manuscript under the sub-heading 'Patient and Public Involvement'.

Response: We have added this statement to the method section, being as follows:

“Patient and Public Involvement

Patients were not involved in the design of this substudy of HATICE. However, patients were involved in the development of the HATICE eHealth application by means of conduction focus groups with the projected target population of the HATICE eHealth application and by means of consulting patient organisations (Dutch Heart Foundation and Dutch and Finnish Alzheimer Association). Results of this substudy were disseminated to the participants by means of a written summary. “

- Please kindly remove the Drawing tools in the main text and use the table tools instead.

Response: we have adjusted this

VERSION 2 – REVIEW

REVIEWER	ANIBAL GARCIA-SEMPERE FISABIO, SPAIN
REVIEW RETURNED	28-Jan-2019

GENERAL COMMENTS	The authors have sufficiently addressed my comments. However, I feel that the manuscript would benefit from a last review of written English before being suitable for publication.
--

REVIEWER	Kate Morton University of Southampton
REVIEW RETURNED	17-Jan-2019

GENERAL COMMENTS	I would like to thank the authors for their thorough responses, and feel the manuscript has been improved following these revisions. I only have one outstanding query to raise which relates to my previous comment; number 15. In response to this comment, the authors explained how the data collection methods were inductive which I agree with. However, my query was about the decision in the results to report the themes for Parts 1 and 2 separately. As it sounds like the whole focus group was analysed together, with data from the discussion around optimal behaviour change support (Part 1) and views of HATICE (Part 2) jointly informing the development of the 3 themes, it makes sense to me that they be reported together, as there seems considerable overlap between e.g. nurses' perceptions of 'Establishing a relationship of trust' in order to change behaviour, and their views on establishing trust for online support to change behaviour. The key recommendations for online support settings could be clearly highlighted in the discussion. However, this is just my personal preference, and as neither of the other reviewers raised it then I am happy if the authors prefer to keep the reporting of Parts 1 and 2 separate. If this is the case, might I suggest that the Results section explains at the start that three themes were developed from the analysis of the focus groups, and these will be reported in relation to Part 1 and 2 separately – to make this more explicit for the reader?
--

VERSION 2 – AUTHOR RESPONSE

Responses to reviewer #2

Reviewer: 2

Reviewer Name: ANIBAL GARCIA-SEMPERE

Institution and Country: FISABIO, SPAIN

Please state any competing interests or state 'None declared': NONE DECLARED

The authors have sufficiently addressed my comments.

However, I feel that the manuscript would benefit from a last review of written English before being suitable for publication.

Response: We followed the recommendation of the reviewer and had the paper reviewed for English writing and grammar by a native English speaking reviewer from our network. Please find the minor textual changes marked throughout the document.

Responses to reviewer #3

Reviewer: 3

Reviewer Name: Kate Morton

Institution and Country: University of Southampton, UK Please state any competing interests or state 'None declared': None declared

I would like to thank the authors for their thorough responses, and feel the manuscript has been improved following these revisions.

I only have one outstanding query to raise which relates to my previous comment; number 15. In response to this comment, the authors explained how the data collection methods were inductive which I agree with. However, my query was about the decision in the results to report the themes for Parts 1 and 2 separately. As it sounds like the whole focus group was analysed together, with data from the discussion around optimal behaviour change support (Part 1) and views of HATICE (Part 2) jointly informing the development of the 3 themes, it makes sense to me that they be reported together, as there seems considerable overlap between e.g. nurses' perceptions of 'Establishing a relationship of trust' in order to change behaviour, and their views on establishing trust for online support to change behaviour. The key recommendations for online support settings could be clearly highlighted in the discussion.

However, this is just my personal preference, and as neither of the other reviewers raised it then I am happy if the authors prefer to keep the reporting of Parts 1 and 2 separate. If this is the case, might I suggest that the Results section explains at the start that three themes were developed from the analysis of the focus groups, and these will be reported in relation to Part 1 and 2 separately – to make this more explicit for the reader?

Response: We understand the reviewer's concerns regarding the presentation of the same themes separately for Part 1 and Part 2. Since the other reviewers did not raise this point, and since we liked to explicitly distinguish the findings originating from the nurses' current clinical experiences and practices (Part 1) from the nurses' recommendations for future online-support (Part 2) we kept with the separation of Part 1 and 2. However, we followed the reviewer's second suggestion to make this more explicit for the reader, by adding a more complete explanation of the Results' structure at the beginning of the Results section:

"We analysed the data from **Part 1** (the nurses' experiences and practices with supporting the process of behaviour change for cardiovascular prevention, including the potential for dementia prevention) (Part 1) and **Part 2** (the nurses' suggestions on how to integrate their experiences in an online-support setting, stimulated by a demonstration of the HATICE-platform²⁶) together, jointly informing the identification of 3 main themes. The themes can be understood as the nurses' preconditions for effective behaviour change guidance in their patients: establishing a relationship of trust; awareness and expectation management; and appropriate timing and monitoring. These were

regarded as being equally important in 'off-line' and 'on-line' health care. Below, they are reported separately in relation to **Parts 1** and **2**, to distinguish the nurses' clinical experiences and practices in current health care settings from their recommendations for optimal online support." (Results, p7-8, lines 37-8)